# An Intelligent Perception Model and Parameters Adjust Method for Quality of Experience

**Yanqin Wu [1], Le Zhang [1], Tiantian Lv [1,\*], Rongrong Guo [1,\*], Liang Xing [2] and Yanchuan Wang [2]**

1   Research Institute of China Telecom Corporation Limited, Beijing 102200, China; wuyq@chinatelecom.cn (Y.W.); zhangle@chinatelecom.cn (L.Z.)
2   China Telecommunications Corporation, Beijing 102200, China; xingliang@chinatelecom.cn (L.X.); wangych@chinatelecom.cn (Y.W.)
\*   Correspondence: lvtt@chinatelecom.cn (T.L.); guorr@chinatelecom.cn (R.G.)

**Abstract:** With the coming of the 5G network era, Quality of Experience (QoE), which is used to describe experience quality of customers in the Direct Connect (DC) network, is critical to operators and hence has been widely studied. The existing evaluation method for QoE is unable to dynamically adjust QoE parameters so there will be deviations to reflect the real experience of customers when complaint events occur. In this paper, a novel progressive QoE evaluation model for network products is formulated. Subsequently, considering the customer complaints occurrences which represent that the real customer experience and previously calculated QoE may mismatch, the rules based on application types to judge whether QoE parameters needs to be adjusted are made. Then, a dynamic adjustment model of QoE parameters is further developed to reflect complaint customer experience. Finally, the obtained experimental result verified that the proposed dynamic perception method for QoE can accurately characterize the real customer experience. The evaluation result for QoE model based on customer satisfaction questionnaires proves the effectiveness of the QoE-aware model. The evaluation result for adjusted QoE model based on business complaints shows that the adjusted QoE calculation variance reduction rate exceeds more than 50%, which indicates that the proposed method has better ability to reflect the real customer experience.

**Keywords:** quality of experience; subjective evaluation; objective evaluation; eXtreme Gradient Boosting

## 1. Introduction

The arrival of 5G promotes the network to enter a new level and has accelerated the development of thousands of industries. Customers' requirements for the continuous improvement of network business have become increasingly high. Operators previously used network performance (e.g., delay, jitter, bandwidth, bit errors, etc.) as the basis for Direct Connect (DC) network business optimization while they ignore customer experience that is a significant factor to enhance core competitiveness. Effectively perceiving and managing customer experience could boost customer satisfaction degree and increase consumer loyalty to the product. As an evaluation index of the quality of customer experience, quality of experience (QoE) is defined by the degree of delight or annoyance of the customer to a service or product [1]. After introducing QoE into the existed complex network environment, how to accurately evaluate QoE is critical to drive service or product optimization with customer experience. Thus, the evaluation of QoE has become a research hotspot in the industry.

Generally, the evaluation of QoE can be divided into two categories: subjective assessment and objective assessment [2]. In the first case, specific assessment processes are needed to be prepared, referred to as subjective tests, while in the second case some mathematical formulas or algorithms are applied, referred to as objective models [3]. Subjective tests are usually based on manageable real life experiments with selected human participants who directly evaluate their experience of an application or service.

These tests need to be thoroughly designed in advance and referred to guidelines and recommendations by standardization bodies. Various evaluation methods may be used for subjective tests. The typical one is that customers score the experience quality based on an absolute rating scale. Besides, Chen et al., proposed a real-time QoE estimation method called the "OneClick" paradigm which requires participants to click a dedicated key whenever he/she feels dissatisfied with the quality of the application in use [4]. The advantage of subjective assessment is that the tests could incorporate and capture any conscious and unconscious aspects of human experience quality evaluation. However, subjective tests are costly, time-consuming and not reproducible on demand [3].

To solve the problem, objective assessment is studied extensively due to measuring the quality perceived by customers without their intervention. Objective assessment are based on technical parameters and mathematical analyses [5]; researchers have built plenty of models to evaluate QoE for a variety of applications (e.g., video streaming, conversational voice, web browsing, file download, etc). Shaikh et al., proposed the linear and exponential functions-based QoE measurements, which are evaluated by experiments [6]. Using the mean opinion score model, Ameigeiras et al., established the mapping relationships between the response time of the web server and the QoE parameter [7]. Alreshoodi and Woods provided generic formulas that parametric QoE models usually follow [8]. Recently, Bouraqia et al., found that the machine learning method can be comprehensively applied more than the mathematical method to measure QoE for streaming services [9]. Besides, Anwar et al., compared K-Nearest Neighbors, Support Vector Machines, Decision Trees and Logistic Regression in terms of perceptual quality to consider the QoE estimation model for Virtual Reality (VR) services [10]. Anchuen et al., summarizes the QoE model characteristics for in four popular multimedia services, YouTube, Facebook, Line, and Web browser, based on area diversity and user diversity and creates QoE model with the ANN Method [11]. Boz, Eren, et al., propose a mobile QoE prediction in the field, which can report descriptive statistics and classification results predicting normal vs. bad QoE in in-the-wild measurements [12]. Paper [13] proposes a general QoE model to facilitate the future 360 VR video streaming mechanism design. Paper [14] proposes three new models to measure the QoE analytically in DASH (Dynamic Adaptive Streaming over HTTP) video services based on the bitrate of the displayed video segments, the PSNR and VMAF of each video segment, respectively. Ting Yue, et al., propose a hybrid network model that integrates a deep neural network (DNN) and an improved recurrent neural network (RNN) for representation learning of view-level QoE evaluation [15]. Ruochen Huang et al., propose data-driven QoE prediction for IPTV service to predict QoE for users of Internet Protocol TV (IPTV) services [16]. H. F. Bermudez et al., evaluate the performance of a live video streaming service over an LTE network in terms of QoE using the ITU-T Rec. P.1203 model in paper [17]. Paper [18] proposes a control loop for transmission power and channel selection, based on Software Defined Networking and Reinforcement Learning (RL), capable of improving Web Quality of Experience metrics, thus benefiting the user. However, the mentioned QoE models where parameters are preset cannot accurately reflect the customer experience of DC network in real-time. Therefore, how to obtain more dynamic and accurate QoE implemented to the DC network business optimization is a key issue to be solved.

The QOE model of the above paper mainly focused on certain network applications facing public customers. In the 5G era, network intelligence and new network-based business has put DC networks and their role in the overall telecommunications network in an unprecedented development perspective. The DC mainly refers to the use of rich wired and wireless network resources to provide guaranteed transmission channels for different group customers to achieve a variety of high-quality information delivery services. The group customers are mainly involved in various fields, such as finance, government, military, agriculture, industry, building etc. The leased-line business for vertical industry has become a new engine for telecom operators' future profit sources. According to diverse transmission network carrying technology, the DC products primarily include Multi-Service

Transport Platform (MSTP)-based DC products, optical transport network (OTN)-based DC products, Internet Protocol Radio Access Network (IPRAN)-based DC products, etc., which could meet the multiple requirements for business scenarios. At present, there are few studies on the customer experience quality for group customer DC.

In view of the above-mentioned facts, the paper proposes a dynamic perception method for QoE to represent real experience of customers. A general perception method of QoE based on progressive calculation model is firstly proposed, which could direct QoE evaluation in different DC network businesses. In this part, two indexes including Experience Range (ER) and Experience Efficiency (EE) are defined to describe the customer experience. Additionally, a novel parametric model is given to calculate the QoE. Considering the customer complaint occurrence which represents that the real customer experience and previously calculated QoE may mismatch, a dynamic adjustment method of QoE parameters is further developed to reflect changed customer experience [19]. Subsequently, abnormal KQIs and corresponding customer characteristic data are pertinently selected. Finally, new adjustment model of QoE parameters is established to calculate the adjusted parameters. Experimental results demonstrate that the proposed QoE evaluation method can dynamically and accurately represent the real customer experience.

The rest of this paper is organized as follows. Section 2.1 introduces perception model for QoE, the calculation process for QoE and the building method of QoE perception model. Section 2.2 introduces the proposed dynamic adjustment method for QoE parameters. Section 3 gives the experimental result based on the Customer Experience Management system. Section 4 is the discussion of the study. Section 5 is the conclusions of the paper.

## 2. Materials and Methods

### 2.1. QoE Perception Model

#### 2.1.1. Introduction of QoE

China Telecom proposed a QoE acquisition method in the patent [20]. Based on the QoE model in the patent and combined with DC network business of China Telecom, this paper proposes a novel five-layer gradual pyramid QoE Indicator model (GPQIM) which defines the correlation among customer experience quality, leased-line product quality, business/service quality and network performance, which is depicted in Figure 1.

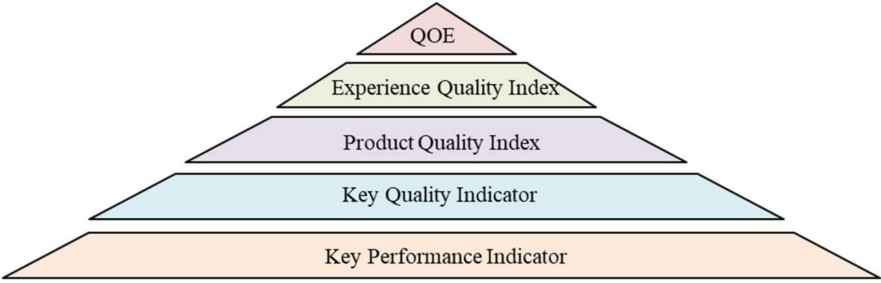

**Figure 1.** Gradual pyramid QoE indicator model.

The selected indicators should be customer-oriented, product-focused, business-related and network-based. Accordingly, the customer experience quality indicators fall into two broad categories: use of comfort giving attention to customer experience of using telecom product and service of comfort placing emphasis on customer experience of receiving telecom service. From the view of product, the above two customer experience quality indicators can be subsequently decomposed to five indicators: availability, stability, provision time-efficiency, assurance time-efficiency and service compliance. Availability refers to the capacity to have access to the DC network whenever it is desired by the customer. Stability means the level of the line network transmission quality. Provision time-efficiency refers to the provision time of line network since the customer ordered the product. Assurance time-efficiency is time of maintaining and optimizing line network offered to the customer. Service compliance is designed to describe customers' subjective

feeling about routine operation service provided by the technician. The above indicators need to be improved and perfected facing different evaluation business.

Figure 2 gives an example of QoE indicator system for VPN-based leased-line product. Eight VPN-related key quality indicators (KQI) and eleven key performance indicators (KPI) are specifically mapped and listed. Different indicators should change with corresponding products as a result of their distinct technical emphasis.

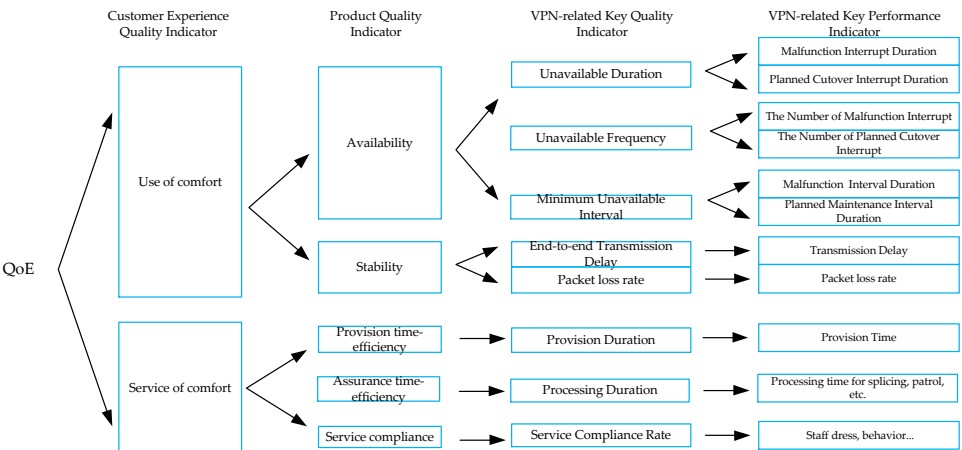

**Figure 2.** Progressive Perception Model for QoE.

### 2.1.2. Calculation Process for QoE

Based on the QoE model in Section 2, the calculation method for QoE is further developed. The flow diagram is shown in Figure 3.

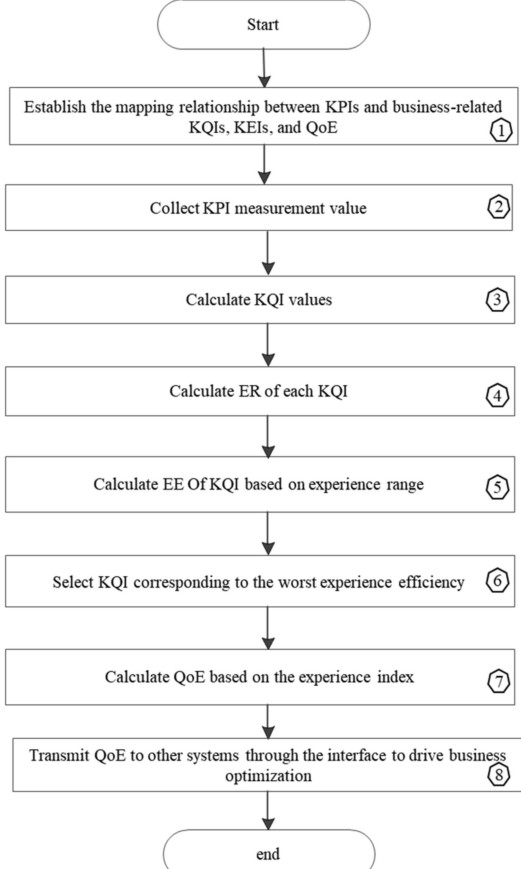

**Figure 3.** Flow chart of proposed progressive perception method for QoE.

Mainly including the following key parts:

(1) Establish the mapping relationship between KPIs and business-related KQIs, KEIs, and QoE.
(2) Collect KPI measurement value.
(3) Calculate the actual measurement values of the KQIs through KPIs. The KPIs and KQIs include multi-dimensional metrics which are shown in Figure 1. For different applications, the KQI metrics could be distinct.
(4) Based on calculated KQIs, the gap between the network business and the customer expectation is defined as ER.
(5) Furthermore, the actual customer perception of the each KQI based on the ER is calculated and named as EE.
(6) Five customer key experience indicators are given in Figure 1, including line network business availability, stability, opening timeliness, assurance timeliness and service comfort. For each indicator, select the worst EE as the measurement of the customer experience because quite a lot of KQIs reflect the same experience indicator.
(7) Then, the overall QoE based on the five indicators is evaluated to perceive the customer experience quality.
(8) Finally, transmit QoE to other systems through the interface to drive business optimization.

KPIs are collected and measured from the external system and the values of KQIs are obtained by mapping the relationship between the KPI and KQI. The customer experience range of KQI is defined in this paper:

$$ER_i = m_i \times \lambda_i / t_i \tag{1}$$

where $i$ denotes $i$th KQI, $m_i$ denotes $i$th KQI measurement, $\lambda$ characterizes the degree of influence of the KQI on the customer, $t$ characterizes the worst degree at which the customer can tolerate the KQI at most. The $\lambda$ and $t$ are set for the customer in advance.

Based on ER for each KQI, historical data and satisfaction survey, the experience efficiency model for each KQI can be obtained by fitting with the Lagrange interpolation method. Figure 4 gives an example of EE fitting curves for "unavailable duration" KQI, and the formula is given by:

$$EE_i = 1 - \frac{ER_i \times 50 - ER_i \times 100 \times (ER_i \times 100 - 50) \times (ER_i \times 100 - 100)}{1.2 \times 10^5} + \frac{ER_i \times (ER_i \times 100 - 50) \times (ER_i \times 100 - 100) \times (ER_i \times 100 - 200)}{2.4 \times 10^7} - \frac{ER_i \times (ER_i \times 100 - 50) \times (ER_i \times 100 - 100) \times (ER_i \times 100 - 200) \times (ER_i \times 100 - 300)}{8.4 \times 10^7} \tag{2}$$

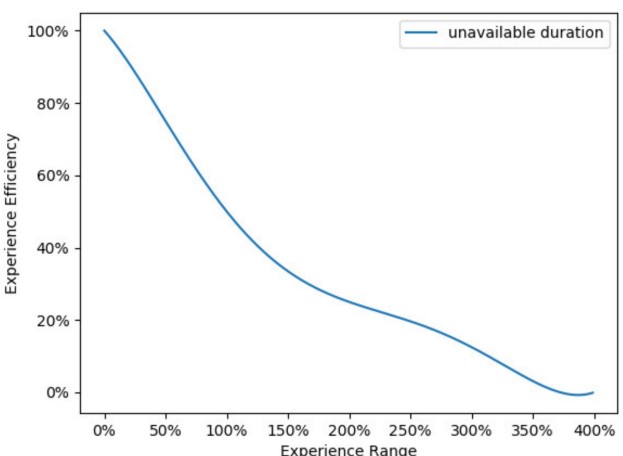

**Figure 4.** Example of EE fitting curves for "unavailable duration".

The proposed EE reflects the customer experience with a specified application. For instance, the value of EE in the range of 100% to 70% indicates the best customer experience.

If 40% is average, 70% to 40% indicates inferior experience. Additionally, 40% to 0% means poor experience.

As Figure 4 shows, the impact of negative factors on customer experience is a non-linear and progressive process. When a factor starts to deteriorate, even if it has not deteriorated to the set expectation value, the customer experience has already started to suffer, and as the deterioration level rises, the quality of customer experience will decline non-linearly.

### 2.1.3. Building Method of QoE Perception Model

For DC network, the key experience indicators include: availability, stability, opening timeliness, assurance timeliness and service comfort. Availability refers to the capacity to have access to line network whenever it is desired by a customer. Stability means the level of the line network transmission quality. Opening timeliness refers to the opening time of line network since the customer order the business. Assurance timeliness is time of maintaining and optimizing offered line network to customers. Lastly, service comfort is the characteristic that routine operation served by the technician do not make customer feel uncomfortable. These five indicators affect and determine the value of QoE. However, there exist some KQIs that characterize the same experience indicator as shown in Figure 5, that is, the mapping relationship between experience indicators and KQI is one to many. When evaluating the quality of an experience indicator, the KQI with the greatest deterioration among all KQIs should be paid most attention. Therefore, the paper selects the worst EE of KQI as corresponding to the experience indicator. Finally, the value of each experience indicator is weighted to obtain the actual QoE of the customer, defined as:

$$QoE = \sum_{i=1}^{5} W_i V_i \qquad (3)$$

where $W_i$ denotes weight of $i$th experience indicator. $V_i$ denotes value of $i$th experience indicator.

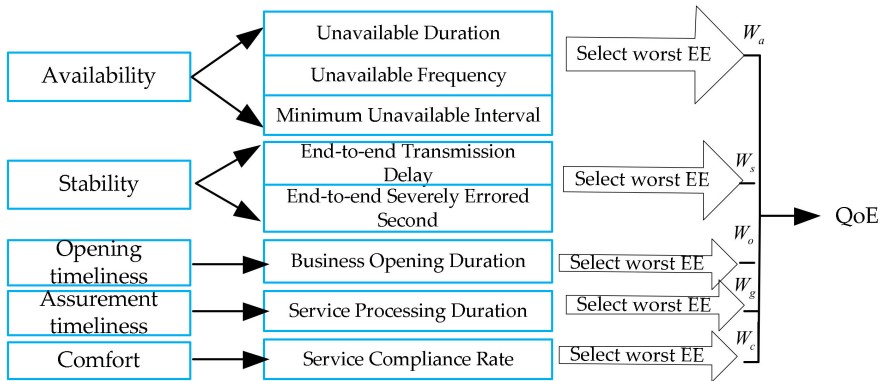

**Figure 5.** Proposed Calculation Method for QoE.

### 2.2. Intelligent Adjustment Method for QoE Parameters

In the process to support the network, businesses have inevitably met customer complaints. When the customer complaints occur, it is important to judge whether the value of QoE needs to be recalculated to correctly reflect real experience quality. The QoE parameters $\lambda$, $t$ and $W$ are generally preset by pre-research or big data analysis in section II, hence should be dynamically adjusted in calculation. The flow chart of the method of dynamic adjustment parameters for QoE is shown in Figure 6.

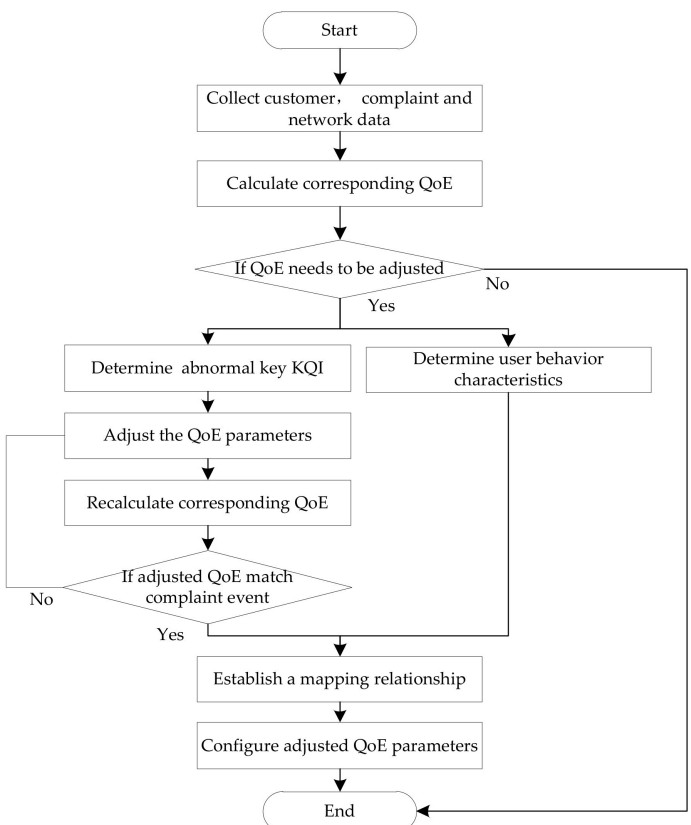

**Figure 6.** Flow chart of proposed intelligent adjustment process for QoE parameters.

The key steps of the proposed method are listed as follows:

(1) Receive customer complaints and collect customer data and network data.
(2) Determine whether the QoE corresponding to the complaints need to be adjusted where the judgement rules are defined based on application types and the XGBoost is exploited to classify the application types of complaints from customers.
(3) Determine the KQIs and customer behavior characteristics associated with the complaints, and optimized model to adjust the QoE parameters is defined.
(4) Then, determine whether the recalculation results match the complaint. If they match, continue the process; otherwise re-adjust.
(5) Finally, establish a mapping relationship between the adjusted QoE parameters and the analyzed customer behavior characteristics.
(6) According to the mapping relationship, when the conditions of customer behavior characteristics are met, the adjusted QoE parameters are configured.

Analyze KPIs associated with the complaints of the same application type and select abnormal ones. Meanwhile, associate customer behavior characteristics which mainly include time period of application used, complaint location, complaint time and DC network type. Then, the most influential KQIs corresponding to the abnormal KPIs are determined with the complaints. To reflect real customer experience, the adjustment model of QoE parameters is given by:

$$p_{new} = \left| \frac{1}{M}\sum_{i=1}^{M} y_i - S \times \sqrt{\frac{1}{N}\sum_{j=1}^{N} \left( p_j - \bar{p_{ori}} \right)^2} \right| \tag{4}$$

where $p$ denotes the QoE parameters need to be adjusted, especially referring to $\lambda$, $t$, $W$ in this paper. $M$ denotes the number of complaints of same application type. $y_i$ denotes the actual values of the KQIs associated with the $i$th complaint event. $N$ denotes the number of

the QoE parameters. $\overline{p}_{ori}$ denotes the average of historical valued of $N$ QoE parameters. $S$ denotes the adjustment coefficient.

Apply the adjusted QoE parameters to recalculate the QoE value corresponding to the complaint: if the QoE value is consistent with the customer experience, the mapping relationship between the new parameters and the customer behavior characteristics should be established. Finally, the QoE parameter configuration rules are set based on the customer behavior characteristics. When calculating the QoE, it is determined whether the customer behavior data matches the parameter configuration rules; if they match, the adjusted QoE parameters are obtained based on the mapping relationship to further calculate QoE value.

## 3. Results

### 3.1. Experiment Data

In the section, experimental test based on the Customer Experience Management (CEM) system developed by China Telecom is conducted to verify the performance of the proposed method. For years, China Telecom has been committed to providing better DC network experience for government and enterprise customers. The network businesses provided mainly include boutique Optical Transmission Net (OTN), Internet Protocol Radio Access Network (IPRAN), Virtual Private Network (VPN) and so on. Based on these special DC, multiple applications are supported and used in companies including office automation, online transaction, videoconferencing, live broadcast, etc.

The log server on the CEM system is developed to collect customer data and network data in real time. Based on the above data, the mapping relationship between time, customer identification, customer data, and network data is established and stored. To validate the progressive perception model for QoE, 1000 customers are randomly selected from CEM system and their score of experiential network quality (out of 100) is collected. Additionally, the variance between the evaluated QoE calculated by perception model and subjective one scored by customers is less than 10. It is stated that the proposed QoE model has the ability to quantify customer experience.

To further verify the dynamic adjustment method for parameters of QoE, complaints are collected when the number is 500. The QoE is evaluated by the proposed progressive perception method. Additionally, the parameters $\lambda$, $t$, $W$ are preset based on expert experience. Additionally, the complaint customers are invited to score the experiential QoE out of 100 according to the absolute rating scale.

### 3.2. Evaluation Result for QoE Model Based on Customers Satisfaction Questionnaires

The element management system is responsible for collecting network data and customer data from network element equipment in real time. The service data are recorded by an external operational support system. Based on the above data, the mapping relationship between time, customer identification, customer data, and network data is established and stored.

To validate the proposed GPQIM and calculation method for QoE, 2000 customers are randomly selected from CEM system and their satisfaction evaluation questionnaires collected to score the experiential network quality according to the absolute rating scale (out of 100). Through dividing the QoE score (0–100) equally into 10 intervals, the ultimate level of customer experience quality is mapped in Table 1.

**Table 1.** Ultimate level of customer experience quality.

| Interval | Level |
| --- | --- |
| (0, 30) | poor |
| (30%, 60) | fair |
| (60%, 80) | good |
| (80%, 100) | excellent |

Figures 7 and 8 give two customer examples about QoE calculation process. The QoE for customers in Figure 7 is fair while in Figure 8 it is excellent. Additionally, there exist some differences in preset weight of product quality. Customer 1 lays more emphasis on availability of the network than stability while customer 2 considers both of them with the same level of importance.

| QOE | | Customer Experience Quality | | | PRODUCT QUALITY | | | VPN-related Key Quality | |
|---|---|---|---|---|---|---|---|---|---|
| value | level | Indicator | value | weight | INDICATOR | VALUE | WEIGHT | Indicator | EE |
| 41.8 | fair | Use of comfort | 33.5% | 80.0% | Availability | 38.2% | 70% | Unavailable Duration | 38.2% |
| | | | | | | | | Unavailable Frequency | 66.02% |
| | | | | | | | | Minimum Unavailable Interval | 85.88% |
| | | | | | Stability | 22.5% | 30% | End-to-end Transmission Delay | 33.4% |
| | | | | | | | | Packet loss rate | 22.5% |
| | | Service of comfort | 75.0% | 20.0% | Provision time-efficiency | 68.8% | 80% | Provision Duration | 68.8% |
| | | | | | Assurance time-efficiency | 100% | 10% | Processing Duration | 100% |
| | | | | | Service compliance | 100% | 10% | Service Compliance Rate | 100% |

**Figure 7.** Calculation process of fair.

| QOE | | Customer Experience Quality | | | PRODUCT QUALITY | | | VPN-related Key Quality | |
|---|---|---|---|---|---|---|---|---|---|
| value | level | Indicator | value | weight | INDICATOR | VALUE | WEIGHT | Indicator | EE |
| 93.6 | excellent | Use of comfort | 92.0% | 80.0% | Availability | 91.6% | 50% | Unavailable Duration | 92.6% |
| | | | | | | | | Unavailable Frequency | 94.9% |
| | | | | | | | | Minimum Unavailable Interval | 91.6% |
| | | | | | Stability | 93.4% | 50% | End-to-end Transmission Delay | 93.4% |
| | | | | | | | | Packet loss rate | 93.7% |
| | | Service of comfort | 100% | 20.0% | Provision time-efficiency | 100% | 80% | Provision Duration | 100% |
| | | | | | Assurance time-efficiency | 100% | 10% | Processing Duration | 100% |
| | | | | | Service compliance | 100% | 10% | Service Compliance Rate | 100% |

**Figure 8.** Calculation process of excellent.

Figure 9 shows the whole QoE results distribution comparison between questionnaires and proposed QoE model. The result shows that the poor, fair, good and excellent experience of objective calculation, which are 0.75%, 1.27%, 11.84%, 86.01%, respectively, are similar to the subjective evaluation, which are 1.10%, 1.88%, 13.32%, 83.70%, respectively, so that the results of the proposed method are reliable.

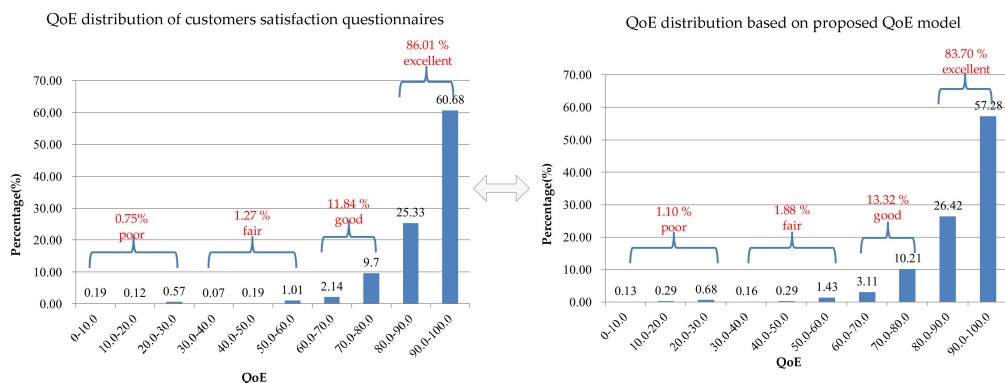

**Figure 9.** Comparison results between questionnaires and proposed QoE model for QoE.

In addition, Figure 10 shows the whole QoE results distribution comparison between questionnaires and other QoE model from reference [11]. The result shows that the poor, fair, good and excellent experience of objective calculation are far from the calculation results of other QOE models, which are 5.90%, 6.50%, 15.20%, 72.40%, respectively, so that the results of the proposed method are reliable.

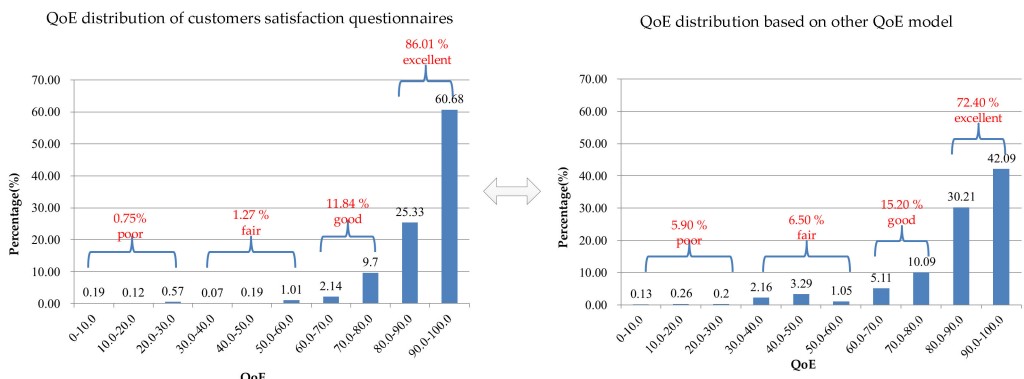

**Figure 10.** Comparison results between questionnaires and other QoE model for QoE.

### 3.3. Evaluation Result for Adjusted QoE Model Based on Business Complaints

According to the classification results, the complaints number of different application types and the variance of deviation between the evaluated QoE calculated by perception model and subjective one scored by customers for each business type are listed in Table 2. It can been seen that there is a huge gap between the evaluated QoE and actual customer experience of the application.

**Table 2.** Compare results between complaints and variance of QoE.

| Application Type | Number of Business Complaints | Variance of QoE |
|---|---|---|
| office automation | 35 | 15.6 |
| online transaction | 1 | 4.3 |
| videoconferencing | 8 | 11.6 |
| live broadcast | 57 | 23.7 |
| overall | 100 | 27.6 |

From Table 2, it is noted that the number of complaint events of online transactions is one, and the number of complaints of office automation, videoconferencing and live broadcast obtained is multiple. Based on the rules of judging whether QoE needs to be adjusted, complaint in office automation is a single incident so the parameters of QoE model for this customer could not be adjusted temporarily. For complaints of the other three application types, the judgment threshold is preset as 80% of the number of complaints.

For each application, because the whole ratio of the number of evaluated QoE that has not yet deteriorated to the number of complaints is greater than the set judgment threshold, the parameter adjustment of QoE model is triggered.

Targeting office automation, video conferencing and live broadcast, identify abnormal KPIs to adjust corresponding QoE parameters. The complaint about the office automation system unavailability as an example is given to illustrate the QoE parameters adjustment process. Additionally, all the complaints are referred to the example to adjust parameters one by one. From Figure 1, the KPIs associated with the complaint are Malfunction Interrupt Duration, Planned Cutover Interrupt Duration, The Number of Malfunction Interrupt Duration, The Number of Planned Cutover Interrupt, Malfunction Interval Duration and Planned Malfunction Interval Duration. According to the mapping relationship, the corresponding KQIs are determined as Unavailable Duration, Unavailable Frequency and Minimum Unavailable Interval, being values of 200 min, 2 and 83 min. Additionally, in the preset QoE parameters, the KQIs thresholds are set to 240 min, 1, 90 min, which are tighter than the actual measured values, hence the calculated QoE according to the perception model does not degrade while the customer experience becomes poor.

Applying Equation (4), the parameters have been adjusted until the variances between subjective QoE scored by customers and adjusted QoE are less than 10. The ultimate adjusted variances and parameters are, respectively, summarized in Tables 3 and 4. Compared between before and after adjustment for variance of QoE, the variance of each application type in Table 4 is less than unadjusted variance in Table 3.

**Table 3.** The adjusted parameters for QoE.

| Parameters | Value before Adjustment | Value after Adjustment |
|:---:|:---:|:---:|
| $\lambda_1$ | 1 | 1 |
| $\lambda_2$ | 0.8 | 0.7 |
| $\lambda_3$ | 0.2 | 0.2 |
| $t_1$ | 240 min | 200 min |
| $t_2$ | 1 | 1 |
| $t_3$ | 90 min | 80 min |
| $W_a$ | 100% | 100% |

**Table 4.** The adjust variances for QoE after adjusting parameters.

| Application Type | Number of Complaints | Variance of QoE after Adjustment |
|:---:|:---:|:---:|
| office automation | 35 | 7.3 |
| online transaction | 1 | 2.1 |
| Videoconferencing | 8 | 5.6 |
| live broadcast | 57 | 9.1 |
| Overall | 100 | 9.8 |

The specific compare results between Tables 2 and 4 are shown in Table 5, and the adjusted QoE calculation variance reduction rate exceeds more than 50%, which indicates the proposed method has better ability to reflect the real customer experience.

**Table 5.** The Comparison between before and after adjustment for variance of QoE.

| Application Type | Variance of QoE before Adjustment | Variance of QoE after Adjustment | Comparison Result before and after Adjustment |
|:---:|:---:|:---:|:---:|
| office automation | 15.6 | 7.3 | 53.2% lower |
| online transaction | 4.3 | 2.1 | 51.2% lower |
| videoconferencing | 11.6 | 5.6 | 51.7% lower |
| live broadcast | 23.7 | 9.1 | 61.6% lower |
| overall | 27.6 | 9.8 | 64.5% lower |

## 4. Discussion

The evaluation result for QoE model based on customers satisfaction questionnaires can prove that the QoE calculation results using the QoE perception model proposed in Section 2.1 are basically consistent with the results of the customer questionnaire, indicating the reliability of the proposed QoE perception model. In order to further improve the calculation accuracy of the QoE model, an intelligent adjustment method for QoE parameters is proposed in Section 2.2. Additionally, the evaluation result for adjusted QoE model based on business complaints can prove the reliability of intelligent adjustment method for QoE parameters.

Even though the intelligent Perception Model and parameters adjust method for Quality of Experience can realize intelligent QoE evaluation without human intervention, it still needs to propose further optimization methods for QoE calculation based on actual service quality and customer survey results, so as to make the calculation results of QoE more accurate and put forward optimization suggestions for some customers with poor and fair QoE calculation degree in Customer Experience Management system to improve customer perception.

## 5. Conclusions

This paper proposes a method to numerically evaluate QoE of DC network business. The progressive model with selected KPIs and KQIs is proposed. Then, aiming at occurrence of complaints, a complete method, including judgement rules of QoE parameters adjustment and QoE parameters adjustment model, is developed to reflect real customer experience as well as optimize network. The evaluation result for QoE model based on customers satisfaction questionnaires proves the effectiveness of the QoE-aware model. The evaluation result for adjusted QoE model based on business complaints shows that the adjusted QoE calculation variance reduction rate exceeds more than 50%, which indicates that the proposed method has better ability to reflect the real customer experience.

## 6. Patents

The authors of this study proposed a QoE acquisition method in the patent [13].

**Author Contributions:** Conceptualization, Y.W. (Yanqin Wu); data curation, R.G. and T.L.; formal analysis, Y.W. (Yanqin Wu) and L.X.; methodology, T.L. and R.G.; software, L.Z.; writing—original draft, T.L. and R.G.; writing—review and editing, T.L., Y.W. (Yanqin Wu), L.Z. and Y.W. (Yanchuan Wang). All authors have read and agreed to the published version of the manuscript.

**Funding:** This work was supported by the research fund of "New generation architecture planning, autonomous system and implementation plan evaluation research and cloud network test platform" [YFL-YWYY-YJY-01-08].

**Data Availability Statement:** Not applicable. Involves confidential data.

**Acknowledgments:** This study acknowledges the China telecom providing experimental materials of business complaints and customer questionnaires of QoE.

**Conflicts of Interest:** The authors declare no conflict of interest.

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
