# Peer review of "An Intelligent Perception Model and Parameters Adjust Method for Quality of Experience"

_electronics, doi:10.3390/electronics11111732_

Round 1

Reviewer 1 Report

This paper attempts to develop a novel progressive QoE evaluation model for network products.
The following are my concerns
1. the novelty of this paper is not clearly demonstrated, especially considering that there is no detailed comparison with state-of-the-art schemes.
2. The references are not sufficient for a journal paper. There are many papers related to QoE modeling, such as  
[r0]. Boz, Eren, et al. "Mobile QoE prediction in the field." Pervasive and Mobile Computing 59 (2019): 101039.
[r1] ”Modeling QoE of Virtual
Reality Video Transmission Over Wireless Networks”, IEEE Global Communica-
tions Conference (GLOBECOM) 2018, Abu Dahabi, UAE, December 2018
[r2] "Point Cloud Video Streaming System: Challenges and Solutions,” IEEE
Network, vol. 35, no. 5, pp. 202-209, September/October 2021 
3. No simulation details are given, and it is difficult to judge the performance of the proposed scheme.

Author Response

Point 1: The novelty of this paper is not clearly demonstrated, especially considering that there is no detailed comparison with state-of-the-art schemes. 

Response 1: Dear reviewer, thanks for your review and comments, and your suggestion is very valuable for this paper. Some contents have been added in section 3.2 to compare the results of the other methods for measuring the QoE. Please review, thanks.

Point 2: The references are not sufficient for a journal paper. There are many papers related to QoE modeling, such as  

[r0]. Boz, Eren, et al. "Mobile QoE prediction in the field." Pervasive and Mobile Computing 59 (2019): 101039.

[r1] ”Modeling QoE of Virtual Reality Video Transmission Over Wireless Networks”, IEEE Global Communications Conference (GLOBECOM) 2018, Abu Dahabi, UAE, December 2018

[r2] "Point Cloud Video Streaming System: Challenges and Solutions,” IEEE Network, vol. 35, no. 5, pp. 202-209, September/October 2021.

Response 2: Dear reviewer, thanks for your review and comments. Some other related refferences have been added in the References part, and the related contents also have been added in section 1.

Please review, thanks.

Point 3: No simulation details are given, and it is difficult to judge the performance of the proposed scheme.

Response 3: Dear reviewer, thanks for your review and comments. Some simulation details of QoE calculation process have been added in section 3.2, including the fair and excellent calculation process based on proposed QOE model. Please review the contents of Figure 5 and Figure 6, thanks.

Reviewer 2 Report

The authors have presented a Quality of Experience QoE evaluation model for network products, which is capable to adjust the QoE parameters. The proposed method is verified by experimental results, which show good agreement.

  • Please explain that if the proposed method is only valid for Direct Connect (DC) networks or can be applied to the other networks, such as mobile networks.
  • Some typos and grammatical errors should be corrected in all of the manuscript parts. For example, in line 318 “… has not been degrade while …”
  • Also, in line 326 it is written that “The specific compare results between Table 3 and Table 4 are shown in Table 5, and …”. The results in Table 5 are obtained by comparing Tables 3 and 4 or Tables 2 and 4? In addition, the variance of QoE after adjustment in Table 4 is “3.3”, while this value is “2.1” in Table 5!
  • It is better to show equation (2) in the mathematical style using fractions (horizontal lines not slash “/”) for more clarification.
  • Kindly explain why quality of service (QoS) is not considered in the proposed model.
  • It is better to compare the results of the other methods for measuring the QoE. This can be reported in the part of the manuscript, where the comparison result between questionnaires and proposed QoE model is shown. This comparison will improve the contribution of the proposed manuscript.

Author Response

Point 1: Please explain that if the proposed method is only valid for Direct Connect (DC) networks or can be applied to the other networks, such as mobile networks.

Response 1: Dear reviewer, thanks for your review and comments. I need to explain thst the proposed macroscopic method, including “Gradual pyramid QoE indicator model”and “Intelligent adjustment method for QoE parameters” also can be applied to the other networks. But for different networks, the indicators are different such as the “Progressive Perception Model for QoE” in Figure2. Because for china telecom, the indicators for DC are sufficient for research and experimentation, so the proposed method is main studied for DC.

Point 2: Some typos and grammatical errors should be corrected in all of the manuscript parts. For example, in line 318 “… has not been degrade while …”.

Response 2: Dear reviewer, thanks for your review and comments. For your suggest, this sentence has been modified, and other similar questions also have been modified. Please review the revised manuscript, thank you.

Point 3: Also, in line 326 it is written that “The specific compare results between Table 3 and Table 4 are shown in Table 5, and …”. The results in Table 5 are obtained by comparing Tables 3 and 4 or Tables 2 and 4? In addition, the variance of QoE after adjustment in Table 4 is “3.3”, while this value is “2.1” in Table 5!

Response 3: Dear reviewer, thanks for your review and comments. For your comments, I have modified this part, including: The results in Table 5 are obtained by comparing Tables 2 and 4; The variance of QoE after adjustment in Table 4 is modofied to “2.1”. I think it's a typo, thanks for your careful review.

Point 4: It is better to show equation (2) in the mathematical style using fractions (horizontal lines not slash “/”) for more clarification.

Response 4: Dear reviewer, thanks for your review and comments. For your comments, I have modified equation (2) used horizontal lines not slash “/”, thanks.

Point 5: Kindly explain why quality of service (QoS) is not considered in the proposed model.

Response 5: Dear reviewer, thanks for your review and comments. Key indicators of Quality of service (QoS) are considered in the PQI, KQI and KPI of the proposed model , including “Availability”, “Transmission Delay”, “Paker loss rate” and so on.

Point 6: It is better to compare the results of the other methods for measuring the QoE. This can be reported in the part of the manuscript, where the comparison result between questionnaires and proposed QoE model is shown. This comparison will improve the contribution of the proposed manuscript.

Response 6: Dear reviewer, thanks for your review and comments, and your suggestion is very valuable for this paper. Some contents have been added in section 3.2 to compare the results of the other methods for measuring the QoE. Please review, thanks.

Round 2

Reviewer 2 Report

The authors have addressed all of my comments and the paper can now  be accepted after considering following minor modifications.

  • In lines 316 and 317, excellent has used two times. Kindly correct this typo.
  • Improve the quality of Figures 5-8 or redraw them as vector-based images.

Author Response

Point 1: In lines 316 and 317, excellent has used two times. Kindly correct this typo.

Response 1: Dear reviewer, thanks for your review and comments. Duplicate “excellent” in lines 316 and 317 also in lines 309 and 310 has been removed. Please review, thanks.

Point 2: Improve the quality of Figures 5-8 or redraw them as vector-based images.

Response 2: Dear reviewer, thanks for your review and comments. The quality of Figures 5-8 has been improved. Please review, thanks.

And the english language and style of the whole paper also have been improved. Please review, thanks.
